# Differentiating Functional Cognitive Disorder from Early Neurodegeneration: A Clinic-Based Study

**DOI:** 10.3390/brainsci11060800

**Published:** 2021-06-17

**Authors:** Harriet A. Ball, Marta Swirski, Margaret Newson, Elizabeth J. Coulthard, Catherine M. Pennington

**Affiliations:** 1Department of Neurology, North Bristol NHS Trust, Bristol BS10 5NB, UK; elizabeth.coulthard@bristol.ac.uk; 2Bristol Medical School, University of Bristol, Bristol BS8 1UD, UK; marta.swirski@bristol.ac.uk; 3Clinical Neuropsychology, North Bristol NHS Trust, Bristol BS10 5NB, UK; Margaret.Newson@nbt.nhs.uk; 4Department of Neurology, NHS Forth Valley, Stirling FK5 4WR, UK; catherine.pennington@ed.ac.uk; 5Centre for Clinical Brain Sciences, University of Edinburgh, Edinburgh EH16 4SB, UK

**Keywords:** functional cognitive disorder, functional neurological disorder, dementia, neurodegeneration, neuropsychometry

## Abstract

Functional cognitive disorder (FCD) is a relatively common cause of cognitive symptoms, characterised by inconsistency between symptoms and observed or self-reported cognitive functioning. We aimed to improve the clinical characterisation of FCD, in particular its differentiation from early neurodegeneration. Two patient cohorts were recruited from a UK-based tertiary cognitive clinic, diagnosed following clinical assessment, investigation and expert multidisciplinary team review: FCD, (*n* = 21), and neurodegenerative Mild Cognitive Impairment (nMCI, *n* = 17). We separately recruited a healthy control group (*n* = 25). All participants completed an assessment battery including: Hopkins Verbal Learning Test-Revised (HVLT-R), Trail Making Test Part B (TMT-B); Depression Anxiety and Stress Scale (DASS) and Minnesota Multiphasic Personality Inventory (MMPI-2RF). In comparison to healthy controls, the FCD and nMCI groups were equally impaired on trail making, immediate recall, and recognition tasks; had equally elevated mood symptoms; showed similar aberration on a range of personality measures; and had similar difficulties on inbuilt performance validity tests. However, participants with FCD performed significantly better than nMCI on HVLT-R delayed free recall and retention (regression coefficient −10.34, *p* = 0.01). Mood, personality and certain cognitive abilities were similarly altered across nMCI and FCD groups. However, those with FCD displayed spared delayed recall and retention, in comparison to impaired immediate recall and recognition. This pattern, which is distinct from that seen in prodromal neurodegeneration, is a marker of internal inconsistency. Differentiating FCD from nMCI is challenging, and the identification of positive neuropsychometric features of FCD is an important contribution to this emerging area of cognitive neurology.

## 1. Introduction

A significant proportion of people attending memory clinics have a functional cause for their cognitive symptoms [1,2,3]. This may be easily recognised by an experienced clinician, but in many cases there is ambiguity, leaving the patient in diagnostic limbo during referral processes, investigations and protracted follow-up. Functional disorders are characterised by an ability to perform a task well at certain times, but with significantly impaired ability at other times (such as limb weakness, or difficulty recalling a number sequence), which often worsens when attention is directed towards the symptoms. This “internal inconsistency” is persistent over time, and does not simply disappear once highlighted to the patient [4].

Data on longer-term outcomes for functional cognitive disorders (FCD) is lacking. A systematic review examining many predominantly memory clinic-based samples with cognitive symptoms suggested functional symptoms to be frequent and that the large majority did not progress to dementia [5]. However, a tertiary memory clinic study found FCD symptoms generally persisted over 20 months’ follow up [6]. A positive diagnosis of FCD is important to facilitate development of evidence based treatments (currently none exist), and also to avoid inappropriate inclusion of people with FCD in studies of neurodegenerative disease [7].

Key to improving FCD diagnosis is the identification of positive clinical features. Specific signs of internal inconsistency have been identified in other functional disorders [8]. Markers of cognitive internal inconsistency include: patient recognition that cognitive performance fluctuates under different circumstances; informant reports of cognitive ability being more favourable than self-report, or a higher level of concern in the subject than their informant (the opposite scenario to that seen in people with neurodegeneration); and performance in clinical interview. For example, providing a fluent, richly illustrated personal history, and/or informant reports of normal cognition [9], is internally inconsistent with self-reported enduring severe cognitive symptoms.

Neuropsychometric assessment allows testing of cognitive sub-domains (such as declarative memory recall versus recognition), in a robust and detailed manner. Deficits in delayed recall and recognition appear to be particularly good markers of amnesia displayed in early Alzheimer’s disease [10,11]. Recognition memory has at times been characterised as indexing familiarity (rather than recollection, which is more closely linked to recall memory). However, the experimental evidence supporting this can alternatively and more parsimoniously be interpreted as distinguishing strong memories from weak memories [12]. Recognition and recall involve overlapping processes located within medial temporal lobe structures, and therefore we might expect different health conditions to affect them differentially. Familiarity requires a subject to identify an object, and judgement that it occurred previously; disorders that affect meta-cognition presumably affect the judgement process more than the identification process.

Psychological assessments commonly include performance validity tests (PVTs), which aim to identify feigned symptoms due to conscious, deliberate underperformance. However, PVTs do not just assess intrinsic motivation: they often require high levels of effort [13]. Test invalidity may be suspected when “easy” items are failed at a higher rate than “hard” items, but the superficially “easier” task (e.g., recognition) may involve a non-completely-overlapping set of cognitive processes compared to the “harder” task (e.g., delayed free recall). PVTs are not consistently failed by people with functional disorders, and the rate of failure appears similar to other clinical conditions such as epilepsy or mild levels of neurodegeneration [13,14], even when there is a strong motivation for participants to do well [15]. The use of PVTs in functional disorders is problematic since these disorders are not caused by conscious deliberate underperformance. However, low scores may suggest poor “cognitive effort” and should lead to careful consideration of the validity of the remainder of a cognitive test battery.

Prior research examining personality traits suggests patients with functional neurological disorders (including motor symptoms, non-epileptic attacks, fibromyalgia and chronic fatigue) may exhibit lower levels of extraversion [16], higher neuroticism [17,18,19,20], higher somatisation, [21] and score highly on items targeting symptom over-reporting [22,23]. Many patients with functional disorders do not have personality difficulties [24] and study results vary depending on the recruitment source, personality model used, and comparator groups [25]. People with functional disorders affecting other organ systems (such as irritable bowel syndrome and fibromyalgia) display a similar profile of cognitive difficulties to one another [13], including difficulties in executive function and selective attention.

The present study examined patterns of mood, cognitive and personality indices amongst a group with FCD, identified from a tertiary memory clinic. We compared them to (a) healthy age-matched controls, and (b) a group with neurodegenerative Mild Cognitive Impairment (nMCI) from the same clinic. This “disease control” group is important in order to reflect the clinical diagnostic differentiation problem. We hypothesised that the nMCI group would have the highest level of cognitive difficulty, and that the FCD group would have greater mood and personality problems. We were careful to collate a representative sample of participants, many of whom experience low levels of anxiety and depression, or may use low levels of medications that can cloud cognitive abilities.

## 2. Materials and Methods

### 2.1. Recruitment

Participants with FCD or nMCI were recruited from a tertiary cognitive disorders clinic, following clinical assessment by a consultant neurologist, appropriate neuroimaging and neuropsychological assessment. All but one attended at least one follow-up appointment. Diagnoses were reviewed at a multidisciplinary meeting comprising three cognitive neurologists, a consultant neuropsychologist and a specialist nurse. Exclusion criteria were: toxic or metabolic causes of cognitive decline (including current or prior alcohol or substance abuse, or significant use of potentially psychoactive medication), major psychiatric disorder, or active systemic disease which could impact on cognition. We purposefully did not exclude persons with mild mood or anxiety symptoms which (in the opinion of the cognitive neurology team) were not the major cause of the reported cognitive symptoms. Patients using low levels of potentially psychoactive medications were included, where clinicians felt this was unlikely to considerably affect cognition. FCD was diagnosed in those with a significant discrepancy between severe self-reported cognitive symptoms and good reported or observed everyday functioning or test performance, no alternative diagnosis, and no evidence of progressive decline. Those with positive clinical evidence of neurodegeneration were excluded from the FCD group. A diagnosis of nMCI was made in participants with evidence of mild impairment on cognitive testing, but with preserved everyday cognitive functioning, and with clinical evidence of neurodegeneration (e.g., abnormal neuroimaging, evidence of a progressive trajectory, or CSF biomarkers, though note these were not all routinely collected for all study participants). One additional participant with nMCI was recruited from the Join Dementia Research database, and their diagnosis was reviewed by a cognitive neurologist (C.P.).

Healthy controls (HC) were recruited from a local database of adult research volunteers and the Join Dementia Research database. Controls self-identified as having no significant cognitive complaints. People with current or prior alcohol or substance abuse, significant use of potentially psychoactive medication, major psychiatric disorder, or active systemic disease which could impact on cognition were excluded.

The project was given Research Ethics Committee approval by the South West—Cornwall and Plymouth Research Ethics Committee, REC reference 15/SW/0298 and IRAS project ID:188539. All participants provided informed consent. The study was funded by the BRACE charity.

A previous publication on this sample [26] describes that during the course of clinic follow-up, the likely diagnostic category changed for two participants, and these were not included in the current analysis (one was re-allocated from nMCI to FCD, and one from FCD to nMCI).

### 2.2. Questionnaires

Participants completed locally-devised questionnaires reporting on their household and employment circumstances, comorbidities and medication use. Comorbidities were listed, and then used to create binary categories indicating:vascular risk factors (hypertension, diabetes mellitus, hypercholesterolaemia, atrial fibrillation, coronary or cerebral or peripheral vascular disease), or use of medications unequivocally for these conditions;functional conditions excluding FCD (irritable bowel syndrome, non-epileptic attack disorder, chronic fatigue syndrome/myalgic encephalomyelitis, fibromyalgia);depression or anxiety, or use of medications unequivocally for this indication (e.g., including SSRIs but excluding amitriptyline and St John’s Wort);medications were listed and screened to identify those that can potentially impair cognition (this comprised patients on low levels of opiates, tramadol, pregabalin, gabapentin, lamotrigine, sleeping tablets, or amitriptyline).

### 2.3. Cognitive and Mood Assessments, and Analysis

Cognitive assessments included the Trail Making Test part B (TMT-B) [27] and the Hopkins Verbal Learning Test-Revised (HVLT-R) [28]. Participants also completed the Depression Anxiety and Stress Scales (DASS) [29]. The DASS subscales, and the TMT-B scores, had skewed residuals (and unequal variances), so were log transformed prior to linear regression analysis. The HVLT data were very skewed, so age-corrected *t*-scores were used in place of the raw data.

In addition to the standard metrics, as a post-hoc analysis examining the HVLT-R word recognition data, we also calculated D-prime, which is a measure of discrepancy between subjects’ ability to distinguish signal from noise [30]. To calculate this, raw probabilities of true hits (H) and false alarms (FA) were generated from the data, then
D prime = z(H) − z(FA)(1)
where z indicates the z-transformation from the probability (note, where probabilities were exactly 1 or 0 they were adjusted away by 0.01). Due to negative skew, the d-prime scores were square transformed prior to linear regression analysis.

### 2.4. Performance Validity, Personality Assessment, and Analysis

The Minnesota Multiphasic Personality Inventory 2 Restructured Form (MMPI-2-RF) [31] produces a number of embedded PVTs as well as 43 personality “substantive scales”. The raw responses for each scale were statistically transformed to uniform *t*-scores.

Using the embedded PVTs, firstly, we looked for participants who consistently gave a pattern of random or fixed responses (predominantly “true” or predominantly “false” responses regardless of the question content, termed VRIN or TRIN). We also identified participants who answered fewer than 90% of the constituent items and “potentially failed” this scale (those whose item failure rate combined with the unanswered items could have put them beyond the failure threshold). Where this pattern of potentially random or fixed responses was found, the remainder of the MMPI-2-RF results for these participants were excluded.

The remaining PVTs identify over or under reporting of symptoms. Over-reporting may occur due to feigning or malingering, but high scores are also generated in the context of multiple somatic symptoms [32]. Therefore, we did not use the remainder of these PVTs to exclude participants’ data, except where a very extreme score was recorded (i.e., an F scale raw score >30; no participants were excluded at this threshold). A similar approach has been used elsewhere [17]. The “over-reporting” scales are: F-r (infrequent reporting), Fp-r (infrequent psychopathology reporting), Fs-r (infrequent somatic reporting) and FBS-r (Fake Bad Scale). Under-reporting (i.e., implausibly virtuous behavior or absence of any bodily symptoms) is measured in scales L-r (uncommon virtues) and K-r (adjustment validity).

To examine the MMPI-2-RF substantive scales, we ran linear regression models on the *t*-scores (which incorporate standardization and controls for age). We excluded individual substantive scales per participant where >10% of items were blank, “unsure” or otherwise not codable. We focused on a small number of pre-defined hypotheses based on published literature [16,17,18,19,21,22,24]. We hypothesized that the FCD group, in comparison to nMCI and HC, would show:higher levels of emotional/internalising dysfunction (“EID”)higher somatic complaints (excluding cognitive complaints) (“RC1”)higher cognitive complaints (“COG”)fewer positive emotional experiences, and avoid social situations and interactions (“INTr”)higher scores on scale “NEGE-r” (indicating those prone to experiencing a wide range of negative emotional experiences, which is associated with neuroticism in the 5-factor model of personality)

Statistical analysis was undertaken using Stata version 15.1.

## 3. Results

### 3.1. Demographics and Background

As previously described [26], participants comprised 21 with FCD, 17 with nMCI and 25 healthy controls. The FCD group was age matched to healthy controls, but the nMCI group was significantly older. There were no group differences in sex nor years of education (Table 1). Rates of employment were 35% in FCD, 6% in nMCI and 48% in healthy controls (chi^2^ = 8.3, *p* = 0.16); note that this is in line with the age difference: those not employed comprised two off-sick, six unemployed and 36 retired. There was no difference in the proportion of participants who live with relatives (85% for FCD, 82% for nMCI and 80% for healthy controls) versus living alone (chi^2^ = 0.19, *p* = 0.91). The percentages reporting being a carer did not differ across groups (25% for FCD, 25% for nMCI and 12% for healthy controls, chi^2^ = 1.58, *p* = 0.45).

Comorbidity and medication data are listed in Table 1. The only statistical difference found was in the level of vascular comorbidity, which was higher amongst the nMCI group (82% had at least one vascular risk factor, compared to 40% in the FCD group and 28% in the healthy controls). This is expected since the nMCI group have a neurodegenerative diagnosis (including cases of vascular MCI), and also higher average age. There was no group difference in the level of other functional disorders (excluding FCD), depression or anxiety, or use of psychoactive medication, but sample sizes are small for binary outcome comparisons. A history of structural brain lesions was uncommon but not absent (FCD: one head injury, one stroke; nMCI: one brain tumour, one stroke; healthy controls: one subarachnoid haemorrhage).

### 3.2. Depression, Anxiety and Stress

Depression, anxiety, and stress levels were each similar across FCD and nMCI groups, but this represented an elevation compared to healthy controls (Table 2).

### 3.3. Cognitive Testing

The raw scores on TMT-B showed those with FCD scoring intermediately between healthy controls and those with nMCI (Table 3). However, this difference disappeared following correction for age in the regression analysis: participants with FCD scored higher (worse) than healthy controls, but no differently to nMCI.

HVLT-R scores for immediate free recall (trials 1–3) were equally impaired in FCD and nMCI groups compared to healthy controls. The same was found for recognition discrimination index (recognition hits minus false positives, and similarly if this was calculated as “D-prime” to better distinguish signal from noise on recognition ability). However, the trial 4 free recall (after a delay of 20 min), and similarly the retention score (trial 4 free recall divided by the best score of trials 2 or 3), showed that participants with FCD scored significantly better than those with nMCI (trial 4 recall 40.3 vs. 31.8, t-2.0, p0.03; retention 43.1 vs. 32.8, t-2.56, *p* = 0.01) (see also Figure 1).

### 3.4. Personality Data

#### 3.4.1. Embedded Performance Validity Measures

First we calculated how many individual items were unscorable (“don’t know”, marked ambiguously, or left blank). We examined how many returned at least 15/338 items unscorable, and those who returned more than one scale unusable because >90% of items in that scale were unscorable (Table 4). The rates were elevated in FCD and nMCI compared to healthy controls (in fact the percentage with high levels of unscorable items was greater in nMCI, 60%, than in FCD, 25%, chi^2^ = 2.2 *p* = 0.04). Next, we identified those who failed or potentially failed the test of fixed or random responding (comprising two FCD participants, four nMCI participants, and no healthy controls). These participants were excluded from further analysis of the MMPI measures.

Both FCD and nMCI groups (compared to healthy controls) scored relatively high on “over-reporting” (this was significantly higher in FCD than healthy controls, *p* < 0.01, but not statistically higher in FCD compared to nMCI, *p* = 0.73). No participant showed significant elevation on scales assessing “under-reporting” of symptoms. Note that we did not exclude data from personality content analysis on the basis of a high rate of unscorable items or over-reporting, unless >10% of items from the scale under examination were unscorable.

#### 3.4.2. Personality Content

This analysis focused on testing 5 pre-defined hypotheses. Patients with FCD scored higher than healthy controls on each of the following scales: EID, RC1, COG and NEGEr. There was no group difference in INTr. However, the FCD group was not significantly different to the nMCI group on any of these indices.

We have also listed the mean and standard errors for each of the substantive scales as an appendix (Appendix A), as this type of personality profile has not been previously published for FCD patients.

## 4. Discussion

This clinic-based comparison of neuropsychological patterns aimed to identify novel markers of FCD. The most striking finding was the similarity between FCD and nMCI groups, in terms of memory performance, levels of depression or anxiety symptoms, and personality traits. Secondly, we found a psychometric pattern suggesting cognitive internal inconsistency in those with FCD. Thirdly, both the FCD and nMCI groups showed a similar rate of sub-optimal performance validity on embedded indices built into the personality assessment. This highlights the importance of carefully considering what “performance validity” really indexes in these populations. The nMCI group were significantly older than those with FCD. Although age alone cannot be used reliably to differentiate neurodegenerative from functional problems in the clinic, older age makes neurodegeneration more likely, so clinicians may be more comfortable making a functional diagnosis in younger patients.

We found a cognitive profile that (if replicated) could be used in positive diagnosis of FCD. People with FCD struggled with immediate recall and recognition tasks, as did those with nMCI, but people with FCD performed surprisingly well on delayed recall. Encoding by FCD participants presumably operated at a reasonable level in order to support their delayed recall, but counter-intuitively, their encoding ability did not translate into results on either the immediate recall or recognition tasks.

One explanation for this cognitive profile follows from the ideas of Yerkes–Dodson “law” (in which increasing arousal, in light of increasing task difficulty, improves task performance but only up to a certain limit, after which performance decreases). Potentially, the delayed recall task matches the FCD group’s preferred optimal level of task difficulty. However, this group elsewhere struggled with other “hard” tasks, such as the National Adult Reading Task (often used as an index of pre-morbid intelligence) [26]. Alongside this difficulty-performance curve, there is also a (small but robust) association between memory performance and memory self-efficacy in healthy people, and such associations are stronger in more attention-demanding tasks (a stronger association when examining delayed recall rather than recognition tasks) [33]. It is plausible that the task that is most cognitively demanding may differ between healthy people and those with FCD or those who are prone to introspection.

An alternative explanation involves the greater requirement for introspection on recognition versus recall tasks (such as incorporation of priors and other metacognitive processes thought to be involved in the generation of internal inconsistency) [34]. A related component contributing to arousal in FCD is likely to be the participant’s perception of their poor performance relative to others. These perceptions may be less tangible during the delayed recall task than in the immediate recall or recognition tasks, which participants might expect healthy people to be able to complete easily. If this were true, then more generally FCD participants should perform well on tasks that rely on implicit memory, but less well on conscious and explicit memory. Better-than-expected performance on delayed recall than immediate recall can sometimes be related to a heavy cognitive or attentional load [35]. This extra cognitive load may be due to a switch from a more automatic to a more explicit mode of processing, as is hypothesised to occur in patients with FCD [13].

It is also important to consider parallels to cognitive impairment in depressed groups, in whom subjective reports of cognitive underperformance and heightened cognitive effort, apparently outweigh objective performance. Although such groups often do not have an objective cognitive score in the impaired range, intra-individual comparisons revealed reductions in performance compared to pre-morbid estimates [36]. Therefore, subjective experience of heightened cognitive effort compared to pre-morbid experiences likely contributes to lack of cognitive confidence, which also “may raise disproportional negative beliefs about one’s ability to function in day-to-day activities.”

Previous studies have shown patients with early Alzheimer’s disease perform poorly on both delayed recall and recognition [10,11]. People with feigned cognitive disorders compared to credible cognitive complainants tested using HVLT-R also showed a different pattern: worse performance in each of delayed recall, retention and recognition [37] and similar results have been found when using the related RAVLT instrument [38,39]. One study has reported RAVLT cognitive patterns in depressive pseudodementia (compared to healthy controls), which showed low scores on almost all subcomponents, with the exception of preserved recognition memory [40]. These findings suggest there are qualitatively different cognitive lapses occurring in participants with FCD, compared to those operating in early neurodegeneration, feigned symptoms, or depression. Caveats to directly comparing these studies include small sample sizes, differing control comparison groups and the mix of different instruments used.

Our personality data firstly allowed scrutiny of the use of embedded symptom validity testing. “Failing” this did not discriminate between functional cognitive problems and difficulties experienced in early neurodegeneration. Secondly, the substantive personality scales were also notable for similarity across these two groups: features linked to internalising problems, somatic complaints and neuroticism were all higher in patients with FCD and those with early neurodegeneration, relative to healthy controls. This raises the possibility that personality differences, as well as scores on self-reporting of mood scales, could be an outcome of cognitive symptoms, whatever their root aetiology.

Important limitations to our study include firstly the small sample size, which limits the robustness of statistical analyses, and may have hindered our ability to detect small between-group differences. For this reason, we curtailed covariates in regression models such that we controlled only for age and sex, and we ran limited analyses on the personality data (testing pre-defined hypotheses as suggested by prior literature). Secondly, the participants have not undergone long-term follow up. However, almost all the FCD and nMCI participants had at least two clinic visits (this resulted in raised diagnostic uncertainty in one participant each of those originally assigned to FCD and nMCI groups, who were excluded for the analyses reported here [26]). Thirdly, the nMCI group was intended to represent a representative clinic sample, with a range of underlying diagnoses (including Alzheimer’s disease and vascular cognitive impairment). Therefore, inclusion in this group was based on consensus clinical opinion, and investigations relevant to their clinical management, rather than a blanket battery of investigations. Fourthly, the cross-sectional nature of this study means that we cannot make inferences about the causal relationship between any of mood, personality or cognitive patterns, and diagnostic group.

## 5. Conclusions

Overall, this study has highlighted similarities on measures of mood, personality and superficial cognitive testing, amongst those with mild cognitive impairments due to functional and neurodegenerative causes. However, detailed analysis of cognitive profiles demonstrated positive evidence of cognitive internal inconsistency in the FCD group. This supports efforts to refine the diagnosis of FCD, away from simply an exclusion of relevant differential diagnoses, towards the detection of positive clinical findings, including neuropsychometric evidence of internal inconsistency.

## Figures and Tables

**Figure 1 brainsci-11-00800-f001:**
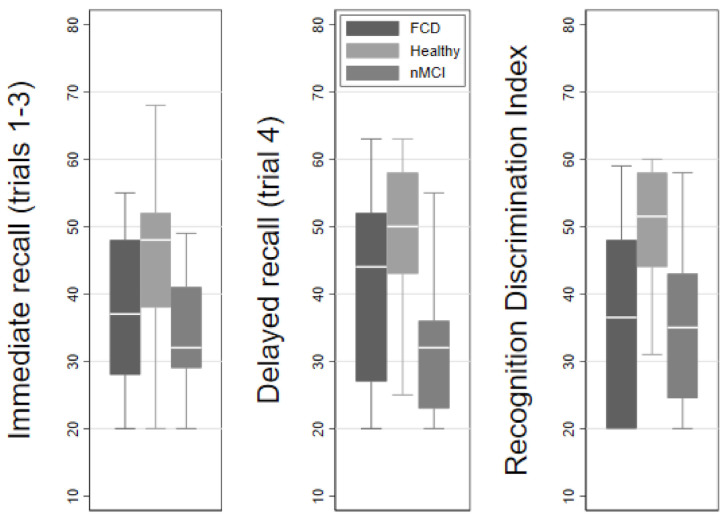
HVLT-R *t*-scores, demonstrating the discriminability of FCD from nMCI and Healthy Control groups, on measures of Immediate Recall, Delayed Recall, and Recognition.

**Table 1 brainsci-11-00800-t001:** Demographics, comorbidities and current medication use per group.

	FCD	Healthy	nMCI	
	Mean (SD)			ANOVA F (p)
Age	58.3 (12.6)	60.8 (5.8)	72.1 (11.7)	9.59 (<0.01)
Years of education	13.8 (2.8)	14.7 (3.7)	14.4 (3.0)	0.50 (0.61)
	*n* (%) per group			Chi^2^ (p)
Sex (female)	10 (48)	18 (72)	8 (47)	3.73 (0.15)
Vascular risk factors	8 (40)	7 (28)	14 (82)	12.55 (<0.01)
Other functional disorder (excluding FCD)	4 ^ (20)	4 ^^ (16)	0 (0)	3.63 (0.16)
Depression or anxiety	5 (24)	3 (12)	3 (18)	1.19 (0.55)
Medication with potential cognitive side effects	9 (43)	8 (32)	5 (29)	1.20 (0.55)

^ 2xfibromyalgia, 1xChronic Fatigue/Myalgic Encephalomyelitis, 1xNon-Epileptic Attack Disorder. ^^ 2xfibromyalgia, 1xChronic Fatigue/Myalgic Encephalomyelitis, 1xIrritable Bowel Syndrome.

**Table 2 brainsci-11-00800-t002:** Depression Anxiety and Stress Scale (DASS) per group.

Depression Anxiety and Stress Scale	Mean (SD)			Regression ^1^: Coefficient (p)	
	FCD	Healthy	nMCI	FCD vs. Healthy	FCD vs. nMCI
Depression	11.1 (11.4)	3.4 (4.8)	8.6 (8.3)	−1.13 (<0.01)	−0.38 (0.30)
Anxiety	9.9 (10.5)	3.1 (4.3)	8 (6.7)	−1.11 (<0.01)	−0.14 (0.71)
Stress	12.2 (9.9)	5.2 (5.1)	11 (8.0)	−0.88 (<0.01)	−0.0.17 (0.62)

^1^ DASS subscales have been log-transformed prior to regression. Regression models control for age and sex.

**Table 3 brainsci-11-00800-t003:** Cognitive testing (TMT-B and HVLT-R).

Scale				Regression: Coefficient (*p*)	
	FCD	H	nMCI	FCD vs. H	FCD vs. nMCI
TMT-B ^1^	Mean (SD)				
Time in seconds	105 (74)	75 (38)	134 (82)	−0.40 (0.02)	−0.02 (0.92)
HVLT-R ^2^	*Raw score mean*; *t*-score mean (SD)				
Total recall (trials 1-3)	*20.9*; 37.3 (12.3)	*24.4*; 45.7 (10.6)	*16.1*; 33.7 (8.8)	9.13 (<0.01)	−3.60 (0.31)
Trial 4 recall (delayed recall)	*7.4*; 40.3 (14.0)	*9.5*; 49.2 (10.0)	*3.8*; 31.8 (10.1)	8.86 (0.02)	−8.52 (0.03)
Retention (trial 4/best of trial 2 and trial 3)	*0.79*; 43.1 (14.1)	*0.92*; 50.0 (9.9)	*0.48*; 32.8 (12.1)	6.99 (0.07)	−10.34 (0.01)
Recognition discrimination index (recognition hits minus false positives)	*7.2*; 36.4 (14.0)	*10.8*; 50.5 (8.3)	*7.6*; 34.9 (11.5)	14.12 (<0.01)	−1.45 (0.71)
	Mean (SD)				
D-prime ^1^	2.22 (1.58)	3.68 (0.84)	2.18 (1.12)	12.62 (<0.01)	−0.30 (0.94)

^1^ TMT-B and D-prime scores were transformed prior to regression. Regression model also controls for age and sex. ^2^ HVLT-R scales (excepting D’) used T-scores incorporating normalisation and age-correction; regression models controlled for sex.

**Table 4 brainsci-11-00800-t004:** Minnesota Multiphasic Personality Inventory (MMPI-2-RF)—validity tests and content scores per group.

	Performance Validity Failure: *n* (Percent)				Personality Content Scales ^5^: *t*-Scores (SD)				
	>15/338 items unscorable ^1^	>1 scale >10% items unscorable ^2^	Potentially random or fixed responding ^3^	Over-report 1+ scale ^4^	EID	RC1	COG	INTRr	NEGEr
FCD (*n* = 20)	5 (25)	8 (40)	2 (10)	11 (52)	59.6 (13.4)	70.6 (18.5)	75.4 (15.2)	61.2 (18.0)	57.6 (15.4)
nMCI (*n* = 15)	9 (60)	9 (60)	4 (27)	6 (35)	58.0 (8.6)	66.6 (12.6)	70.7 (12.8)	60.2 (11.5)	55.5 (13.2)
Healthy (*n* = 25)	0 (0)	1 (4)	0 (0)	4 (16)	50.1 (11.7)	57.2 (10.3)	55.7 (13.7)	55.8 (13.5)	48.4 (9.91)
	Group comparison: Chi ^2^ (*p*)				Group comparison: coefficient (*p*)				
FCD vs. nMCI	4.4 (0.04)	1.4 (0.24)	1.7 (0.20)	0.12 (0.73)	1.67 (0.73)	0.68 (0.91)	−3.30 (0.59)	2.34 (0.71)	1.57 (0.76)
FCD vs. Healthy	7.0 (0.01)	9.00 (<0.01)	2.6 (0.11)	9.4 (<0.01)	−10.16 (0.01)	−12.35 (0.01)	−21.08 (<0.01)	−3.98 (0.42)	−9.40 (0.03)

^1^ Out of 338 items in MMPI-2-RF. “Unscorable” means left blank, or marked “cannot say” or marked both true and false for one item. ^2^ Out of 50 scales. ^3^ Random responding (VRIN) indicates variable-inconsistent responses to items of consistent content. Fixed responding (TRIN) indicates fixed True or False responses regardless of items content. ^4^ Over-reporting scales comprised: F-r (infrequent reporting), Fp-r (infrequent psychopathology reporting), Fs-r (infrequent somatic reporting) and FBS-r (Fake Bad Scale). ^5^ Individual scales were excluded from further analysis if either the individual scale had >10% unscorable items; or the participant had potentially random or fixed reporting. EID = Emotional/Internalising Dysfunction; RC1 = Somatic complaints (note this does not include any cognitive complaints); COG = cognitive complaints; INTRr = a scale denoting “fewer positive emotional experiences, and avoidance of social situations and interactions”; NEGEr = a scale denoting being “prone to experiencing a wide range of negative emotional experiences”, this is associated with Neuroticism in the 5-factor model of personality.

## Data Availability

Anonymized summaries of the data presented in this study are available on request, following suitable application to the corresponding author. The data are not publicly available due to privacy and informed consent reasons.

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
