# Peer review of "Differentiating Functional Cognitive Disorder from Early Neurodegeneration: A Clinic-Based Study"

_brainsci, 2021, doi:10.3390/brainsci11060800_

Round 1

Reviewer 1 Report

The authors provide an interesting view on the topic. It has good supporting evidence and it is of interest for the scientific community. Please take in consideration the comments provided.

Abstract

L25: Comparison between t values is a little bit useless. Please, use regression coefficients or other effect size statistic to perform this comparison. 

Introduction 

L36:”People presenting to memory clinics” it's an awkward term and may lead to confusion. 

L66-L70: I understand the main idea but it’s not well expressed. 

Hypothesis not included in the introduction

  1. Materials and Method

2.1 Recruitment

L109: Please, include the complete list of diagnostic criteria

L111: Why was just one participant recruited from the Join Dementia Research database?

L115: Although I do not doubt the cognitive neurology team’s opinion, this methodological decision requires more than that. Please, include some references which justify that anxiety symptoms do not impact on cognition.  

L119-112: If these are inclusion and exclusion criteria, please, list them in the paragraph above.

L123: Please, list and differentiate exclusion and inclusion criteria for nMCI sample.

L129: Please, list and differentiate exclusion and inclusion criteria for control sample.

2.2 Questionnaires

L136: Please, if some of these questionnaires are published, they should be referenced.

L146: Please, include the complete list of medication which can potentially impair cognition, as you consider. Regarding participants who went under this kind of medication, were they excluded from the final sample?

L150: Was there a criteria for selecting those cognitive tests? 

2.3 Cognitive and mood assessments, and analysis

L149: Sections 2.3 and 2.4 should be reorganized in 2.3 Assessment and 2.4 Statistical analyses. All statistical analyses should be included in just one section in order to facilitate the reader's understanding. 

L162: Why square transformed?

2.4 Performance validity, personality and analysis

L169-176: Please, clarify if this procedure follows the MMPI technical manual’s procedure or any other published procedure.

Results

Regarding the whole Results section, it is essential to describe the procedure followed by the authors in order to control for multiple comparisons. Several statistical tests are performed increasing Type I error rate. Thus, control for multiplicity is essential to sustain the results.

L203: Please, provide justification for sample sizes (probably this justification suits better in Participants section). Is sample size justified because of participants’ availability and accessibility or as a result of statistical procedures such as a priori power analysis, precision analysis, etc?

L205: Please, include standard deviations for age in the three samples and the corresponding effect size for the ANOVA.

L206: Please, include the number/percentage of men/women per sample.

L207: Please, include means and standard deviations per sample in years of education

L214-217: This paragraph does not describe statistical results, hence, it belongs to the Method section.

L238: Description of regression results should include regression coefficient

L239: Does regression models control for age and gender because these variables are included in the models or because t-scores are used instead of raw scores?

L248: Same as for L238

L305-308: Please, include effect sizes for this statistical tests.

Discussion

L323-325: Very interesting point 

L327: Although your results are very interesting, I would not say that a sample size of 21 FCD participants is a cognitive profile. I would change that sentence. You could suggest that these results could lead to a differentiable cognitive profile, for example. 

Others

L428: Please, include a statement about availability of the Join Dementia database.

Author Response

Reviewer 1

Abstract

L25: Comparison between t values is a little bit useless. Please, use regression coefficients or other effect size statistic to perform this comparison. 

Thanks for this comment, the value has been changed to the relevant coefficient in the abstract.

Introduction 

L36:”People presenting to memory clinics” it's an awkward term and may lead to confusion. 

We have changed this to “people attending memory clinics”.

L66-L70: I understand the main idea but it’s not well expressed. 

Thank for this comment, we have slightly re-worded to “However, the experimental evidence supporting this can alternatively and more parsimoniously be interpreted as distinguishing strong memories from weak memories”.

Hypothesis not included in the introduction

We have re-worded the last paragraph of the Introduction to make the hypotheses clearer.

 Materials and Method

2.1 Recruitment

L109: Please, include the complete list of diagnostic criteria

We have merged the first few paragraphs of 2.1 to make it clear how diagnoses were made, and exclusions.

L111: Why was just one participant recruited from the Join Dementia Research database?

Just one participant was recruited from JDR into the nMCI group, since we were able to recruit nearly all participants from our cognitive clinic.

L115: Although I do not doubt the cognitive neurology team’s opinion, this methodological decision requires more than that. Please, include some references which justify that anxiety symptoms do not impact on cognition.

We do not disagree that anxiety symptoms impact upon cognition. We have re-worded the sentence to better make our point: “We purposefully did not exclude persons with mild mood or anxiety symptoms which (in the opinion of the cognitive neurology team) were not the major cause of the re-ported cognitive symptoms“. Please also note the final sentence of the Introduction, where we explain our aim of collating a representative sample, as many patients we encounter with FCD have a low level of anxiety or mood symptoms.

L119-112: If these are inclusion and exclusion criteria, please, list them in the paragraph above.

Thank you, this has been added into the first paragraph of section 2.1.

L123: Please, list and differentiate exclusion and inclusion criteria for nMCI sample.

Thank you, this has been added into the first paragraph of section 2.1.

L129: Please, list and differentiate exclusion and inclusion criteria for control sample.

 Thank you, this has been added in lines L132 -134.

2.2 Questionnaires

L136: Please, if some of these questionnaires are published, they should be referenced.

These were not published questionnaires; this has now been made clear: “Participants completed locally-devised questionnaires reporting on their house-hold and employment circumstances, comorbidities and medication use”.

L146: Please, include the complete list of medication which can potentially impair cognition, as you consider. Regarding participants who went under this kind of medication, were they excluded from the final sample?

All medications were screened for potential to impair cognition; this sentence has now been clarified to include the full list of medications thus identified: “Medications were listed and screened to identify those that can potentially impair cognition (this comprised patients on low levels of opiates, tramadol, pregabalin, gabapentin, lamotrigine, sleeping tablets, or amitriptyline).”

Please refer to the final sentence of the introduction, where we explain we are aiming at a representative sample, and many such patients we encounter are on low levels of these types of medications. It is stated in the first paragraph of 2.1 that we did not exclude these participants.

L150: Was there a criteria for selecting those cognitive tests? 

There was no systematic process of selection of these tests; clinical judgement was used to identify appropriate tests, in order to interrogate the hypotheses that have come from prior literature.

2.3 Cognitive and mood assessments, and analysis

L149: Sections 2.3 and 2.4 should be reorganized in 2.3 Assessment and 2.4 Statistical analyses. All statistical analyses should be included in just one section in order to facilitate the reader's understanding. 

We take the reviewer’s point, but we feel these sections are best kept as they are, as these sections are mainly dealing with assessment (and the brief explanations of analysis are best understood alongside the scales assessed and their defining acronyms; the readers of which could otherwise be unfamiliar with all the acronyms).

L162: Why square transformed?

We have expanded the sentence to make this clear: “Due to negative skew, the d-prime scores were square transformed prior to statistical analysis.” We have also clarified via the headers and footnotes in Table 3.

2.4 Performance validity, personality and analysis

L169-176: Please, clarify if this procedure follows the MMPI technical manual’s procedure or any other published procedure.

We did not follow the MMPI technical manual exclusions, and we are clear in section 2.4 where we have examined such indices but not used them for the purposes of exclusion. We did follow the protocol for VRIN and TRIN; we took note of over-reporting but didn’t use it to exclude participants since that does not adequately account for people with functional symptoms; and we used each scale only if there are 90%+ valid items for that scale. We also provided a reference where a similar approach has been used (Cragar et al 2005).

Results

Regarding the whole Results section, it is essential to describe the procedure followed by the authors in order to control for multiple comparisons. Several statistical tests are performed increasing Type I error rate. Thus, control for multiplicity is essential to sustain the results.

We did not adjust for multiple comparisons, since these are exploratory analyses testing separate hypotheses. We were mindful of the multiple testing problem in that we only assessed 5 of the very numerous personality indices generated by the MMPI, in order to focus on hypotheses as indicated by the relevant literature.

L203: Please, provide justification for sample sizes (probably this justification suits better in Participants section). Is sample size justified because of participants’ availability and accessibility or as a result of statistical procedures such as a priori power analysis, precision analysis, etc?

We agree our sample size is small, and we have highlighted this in the manuscript, as well as the  group sizes in the abstract. Since these were exploratory analyses, formal power calculations were not possible. Nonetheless, our results had sufficient power to detect various differences between FCD and normal controls, whereas FCD and nMCI were much more alike.

Although the sample size is small, it is the largest sample assessing cognitive, affective and neuropsychometric characteristics of FCD, of which we are aware.

L205: Please, include standard deviations for age in the three samples and the corresponding effect size for the ANOVA.

We have moved these details from the text of the Results section, to Table 1, to facilitate display of standard deviation/ percentages etc.

L206: Please, include the number/percentage of men/women per sample.

We have moved these details from the text of the Results section, to Table 1, to facilitate display of standard deviation/ percentages etc.

L207: Please, include means and standard deviations per sample in years of education

We have moved these details from the text of the Results section, to Table 1, to facilitate display of standard deviation/ percentages etc.

L214-217: This paragraph does not describe statistical results, hence, it belongs to the Method section.

Thank you for this comment, we have moved it to be the final paragraph of section 2.1.

L238: Description of regression results should include regression coefficient

We have now included these in Table 4.

L239: Does regression models control for age and gender because these variables are included in the models or because t-scores are used instead of raw scores?

In Table 2 (DASS), no T scores were used, the regression models controlled for age and sex. This is explained in the Method section; we have made this more clear via linking to the footnote beneath the Table.

L248: Same as for L238#

We have now included these in Table 4.

L305-308: Please, include effect sizes for this statistical tests.

We have now included these in Table 4. 

Discussion

L323-325: Very interesting point 

L327: Although your results are very interesting, I would not say that a sample size of 21 FCD participants is a cognitive profile. I would change that sentence. You could suggest that these results could lead to a differentiable cognitive profile, for example. 

We have changed the relevant sentence to “We found a cognitive profile that (if replicated) could be used in positive diagnosis of FCD”.

Others

L428: Please, include a statement about availability of the Join Dementia database.

Join Dementia Research: Join Dementia Research (https://www.joindementiaresearch.nihr.ac.uk) is a UK based database of people interested in volunteering for ethically approved UK based dementia research studies. We have added a statement regarding this into the acknowledgements.

Reviewer 2 Report

In this study, the authors aimed to improve the clinical characterisation of Functional Cognitive Disorder (FCD), in particular its differentiation from early neurodegeneration. I have the following concerns regarding this manuscript.

  • The topic reported is absolutely interesting. Nevertheless, the data is mainly collected from a small sample size of participants, which is an important limitation to the study. I am afraid of an underpowered estimation from the current small scale study. What is the power of your study?
  • Participants were comprised of 21 with FCD, 17 with nMCI, and 25 healthy controls and described in section 2.1. I strongly suggest adding a figure showing the enrollment selection flow chart for the study population to present nicely your inclusion/exclusion process.
  • All subjects included in the survey completed questionnaires. The results from asking the participants may not be so reliable. Keep asking the participants a series of questionnaires may get a random distribution of percentage. Please discuss.
  • A table demonstrating the demographic characteristics of the participants is missing.
  • In table 4, FCD (n=20) and nMCI (n=15), why n not 21 and 17?
  • Functional Cognitive Disorder was used many times in the text. It should be expanded in its first use. In subsequent used, it should be abbreviated.
  • Differentiating FCD from nMCI is challenging. Tell the readers more of the importance of differentiating FCD from nMCI.
  • According to the regulations of submission of Brain Sciences: In the text, reference numbers should be placed in square brackets [ ], and placed before the punctuation. Please be consistent throughout the text.

Author Response

Reviewer 2

  • The topic reported is absolutely interesting. Nevertheless, the data is mainly collected from a small sample size of participants, which is an important limitation to the study. I am afraid of an underpowered estimation from the current small scale study. What is the power of your study?

We agree our sample size is small, and we have highlighted this in the manuscript, as well as the  group sizes in the abstract. Since these were exploratory analyses, formal power calculations were not possible. Nonetheless, our results had sufficient power to detect various differences between FCD and normal controls, whereas FCD and nMCI were much more alike.

Although the sample size is small, it is the largest sample assessing cognitive, affective and neuropsychometric characteristics of FCD of which we are aware.

  • Participants were comprised of 21 with FCD, 17 with nMCI, and 25 healthy controls and described in section 2.1. I strongly suggest adding a figure showing the enrollment selection flow chart for the study population to present nicely your inclusion/exclusion process.

We apologise this was not clear; we have altered the recruitment section to address Reviewer 1’s concerns and on balance felt a diagram would not add to the text description.

  • All subjects included in the survey completed questionnaires. The results from asking the participants may not be so reliable. Keep asking the participants a series of questionnaires may get a random distribution of percentage. Please discuss.

A questionnaire is vulnerable to inaccurate reporting by the participant, and this is an important point. We excluded from the analyse participants who showed a trend of fixed or random responding on the MMPI, or had an excessively high proportion of unscorable items (L178-185).

  • A table demonstrating the demographic characteristics of the participants is missing.

Please see the added rows at the top of Table 1, which hopefully makes it easier for the reader to find (this information was previously in the text of the Results section).

  • In table 4, FCD (n=20) and nMCI (n=15), why n not 21 and 17?

This was due to not all participants completing all of the assessments.

  • Functional Cognitive Disorder was used many times in the text. It should be expanded in its first use. In subsequent used, it should be abbreviated.

We have replaced the instances of this, except the first mention in the abstract and the main text.

  • Differentiating FCD from nMCI is challenging. Tell the readers more of the importance of differentiating FCD from nMCI.

We have expanded a sentence in the first paragraph “This may be easily recognised by an experienced clinician, but in many cases there is ambiguity, leaving the patient in diagnostic limbo during referral processes, investigations and protracted follow-up”. We also feel that the results of this paper highlight superficial similarity between FCD and nMCI, as discussed in the second sentence of the Discussion. Making this distinction is important to give a correct diagnosis, prognosis and management plan based on each. 

  • According to the regulations of submission of Brain Sciences: In the text, reference numbers should be placed in square brackets [ ], and placed before the punctuation. Please be consistent throughout the text.

Thank you for pointing out our inconsistency, this has now been rectified.

Reviewer 3 Report

This study compares and differentiates the clinical characterization of Functional Cognitive Disorder from neurodegenerative Mild Cognitive. The identification of positive neuropsychometric features of FCD will greatly help clinicians diagnose and provide therapy for FCD patients. This paper was well written. My only question is:

As nMCL group (72.1) is much older than FCD group (58.3), is aging possibly a key factor for the observed testing differences between these two groups?

Author Response

Reviewer 3

My only question is:

As nMCL group (72.1) is much older than FCD group (58.3), is aging possibly a key factor for the observed testing differences between these two groups?

We agree that ageing is possibly a key factor, which is why we ensured age is controlled for in all subsequent analyses.